# Effects of Strong Ground Motion with Identical Response Spectra and Different Duration on Pile Support Mechanism and Seismic Resistance of Spherical Gas Holders on Soft Ground

**Mio Kobayashi** [1,*], **Toshihiro Noda** [2] , **Kentaro Nakai** [2] , **Toshihiro Takaine** [3] **and Akira Asaoka** [4]

1 Tokyo Gas Co., Ltd., 1-5-20 Kaigan, Minato-ku, Tokyo 105-8527, Japan
2 Department of Civil and Environmental Engineering, Nagoya University, Furo-cho, Chikusa-ku, Nagoya 464-8603, Japan; noda@civil.nagoya-u.ac.jp (T.N.); nakai@civil.nagoya-u.ac.jp (K.N.)
3 GEOASIA Research Society, Furo-cho, Chikusa-ku, Nagoya 464-8603, Japan; takaine@geoasia.jp
4 Association for the Development of Earthquake Prediction, 1-5-18 Sarugaku-cho, Chiyoda-ku, Tokyo 101-0064, Japan; asaoka@adep.jp
* Correspondence: mio@tokyo-gas.co.jp; Tel.: +81-3-5400-7567

**Abstract:** Safety measures are required for spherical gas holders to prevent them from malfunctioning even after a large earthquake. In this study, considering the strong nonlinearity of the ground and damage to the pile during an earthquake, a three-dimensional seismic response analysis of the holder–pile–ground interaction system was conducted for an actual gas holder on the soft ground consisting of alternating layers of sand and clay. In the analysis, the seismic response of the structure to strong ground motions of different durations with the same acceleration response spectrum was verified. The results show that the piles were relatively effective in controlling the settlement when the duration of the earthquake motion was long. This is because the axial force acting on the pile increased due to the redistribution of the holder load caused by the lowering of the effective confining pressure of the sand and clay layers during the earthquake, which increased the bearing capacity of the pile. In contrast, when the duration of the seismic motion was short, the piles had little effect on the reduction in the settlement because the maximum acceleration was higher than that in the former case, and the piles immediately lost their support function.

**Keywords:** seismic resistance; earthquake duration; response spectra; soil–water coupled finite deformation analysis; gas holder; pile foundation; ultra-soft ground; national resilience

## 1. Introduction

Spherical gas holders (henceforth, "holders") in Japanese gas companies have been constructed since the 1950s, and there have been no confirmed cases of damage caused by earthquakes. Even so, the holders are regarded as important industrial facilities, and there is a need to advance efforts to ensure thorough safety measures that prevent malfunctioning even in the event of a large-scale natural disaster such as a Nankai Trough earthquake with a magnitude of 8 to 9 or a Tokyo Inland earthquake with a magnitude of approximately 7 [1,2]. Therefore, these holders must be verified to be able to withstand collapse during level-2 (L2) strong ground motion earthquakes. It is considered effective with regards to the deformation performance of the pile foundation and ground to verify the seismic resistance of the coupled holder–pile–ground system, which considers the non-linear response of the ground during strong earthquakes and the damage limit of the pile based on this.

Regarding this, Boulanger et al. [3], Turner et al. [4], and Nogami et al. [5] for example, proposed analysis models of the mechanical interaction between the pile and ground. However, none of these studies focused on the strong non-linearity of the pile and ground at the damage limit level. Miyamoto [6], Kimura et al. [7,8] and Kaneko et al. [9] showed that ground deformation and the pile foundation response in the liquefied ground could be reproduced using elasto-plastic finite element analysis; however, there were still issues with

the evaluation of the strong non-linearity of the ground. Furthermore, Tombari et al. [10] showed that the non-linear seismic response of the pile was not significantly affected by the seismic motion duration, and the results fell within the category that could be modeled with an equivalent linearization method. However, this study did not address the modeling of the strong non-linearity of the ground.

With these points in mind, the authors evaluated the holders installed in alluvial (or Holocene) lowland areas in the southern Kanto region, which had alternating layers of loose sand and soft clay, by conducting three-dimensional soil–water coupled finite deformation analyses of an integrated holder–pile–ground system [11] (SYS Cam-clay model [12] was employed as the elasto-plastic constitutive equation of the soil skeleton), as well as seismic resistance verifications [13,14] during L2 seismic motion (strong ground motion) according to the Japan Gas Association "Seismic design guidelines for manufacturing equipment [15] (henceforth, "JGA guidelines"). The results showed that post-earthquake consolidation settlement occurred in association with the disturbance of the clay layer (i.e., decrease in rigidity) in addition to the liquefaction of the sandy layer due to the large acceleration of the input seismic motion. However, it was confirmed that there was no uneven settlement or significant deformation of the holder that would lead to service problems (i.e., continuous use) after the earthquake. Additionally, all the piles were damaged in several places, and all support functions were almost completely lost immediately after the earthquake; therefore, it was also shown that the amount of settlement of the holder during and after the earthquake was almost equivalent to the case when the pile was not assumed to be present. However, these evaluations set input seismic motions with a short duration corresponding to near-field earthquakes according to the JGA guidelines, which determined the seismic motion according to the installation point of the holder. Therefore, there have been no evaluations of strong ground motions with a long duration. Moreover, there were vibrations lasting over five minutes in the Kanto Plain, where the holder was located, despite being over 300 km from the epicenter off the Pacific coast of the Tohoku earthquake in 2011 [16]; this has been indicated as a cause for the liquefaction of the ground in coastal areas, which contain large amounts of fine particles [17]. There are also concerns regarding long-duration vibrations from the Nankai Trough megathrust earthquake, which is expected to occur in the future. Therefore, there is also a need to determine the effects of long-duration earthquakes on holders. In fact, studies on long-duration earthquakes have already begun in some industrial facilities [18]. For example, there were severe damages to oil storage tanks due to liquid sloshing during the 2003 Tokachi-oki earthquake, which was caused by long-period and long-duration seismic waves [19]. Additionally, the importance of considering the detailed structures of the soft near-surface sediments was recognized [20].

In this paper, a three-dimensional soil–water coupled finite deformation analysis was conducted for the holder of the previous study [13] using long-duration strong ground motions, and a seismic performance evaluation was performed. The input earthquake motion used in the analysis is the trench-type earthquake motion specified in the JGA guidelines. In order to compare the results with the results of previous studies (i.e., analysis results of short duration strong ground motion) [13,14], the amplitude of the acceleration response spectrum was adjusted to be the same as that of the strong ground motion set in the previous study. In addition, it was discussed that the effects of input seismic motion characteristics such as duration and maximum acceleration on the pile's damage extent and support functions from the perspectives of non-linear interactions between an alternating sand–clay layer ground. The analysis code used in the present study was the same as that in a previous study [11], which allows the use of the SYS Cam-clay model [12] to conduct evaluations from the sand to the intermediate soil and clay in the ground as well as handle the mechanical response of the coupled holder–pile–ground system during and after the earthquake.

This study aims to contribute to one of the crucial Sustainable Development Goals (SDGs) adopted in the UN agenda, "11: Make cities and human settlements inclusive, safe, resilient, and sustainable". The holders are essential for the stable supply of city gas.

By utilizing the seismic verification carried out in this study, the holders can continue to function even after a large-scale natural disaster, thereby minimizing the damage to the economy and society without causing catastrophic damage, as is the case with earthquake countermeasures for industrial lifeline and equipment [21].

## 2. Analysis Conditions

The analysis conditions for the ground and holder in this study were identical to those of the previous study [13], with the exception of the input seismic motion. Therefore, only the main points of the analysis conditions are described below.

### 2.1. Ground Modeling

Figure 1 [14] shows the soil boring log of the holder installation ground and the one-dimensional finite element mesh used in the analysis. The first two meters of the surface layer was sandy buried soil (F layer), which was approximately six meters of the alluvial (or Holocene) sand layer (Yus layer, GL −2 m–8 m) and approximately 14 m of the alluvial (or Holocene) clay layer (Ylc layer, GL −9 m–23 m). A hard diluvium (or Pleistocene layer) with a large N value was deposited in the deep layer. The F and Yus layers have low liquefaction resistance, which indicates a high risk of liquefaction. The Ylc was a layer in the soft state because it was not only mainly silt but also had an N value of almost zero, and there were concerns that the layer, despite being clayey soil, would decrease in rigidity owing to large vibrations [22]. Figure 1c shows a finite element mesh for 1D analysis to characterize the seismic response of the ground, as mentioned later. Figure 2 shows the three-dimensional finite element mesh of the ground and holder used in the analysis. The fine sand To-s2 layer in Figure 1a was set as the engineering bedrock, and the upper side was analyzed in this study. The foundation was set as a horizontal layer, with the surrounding boring data as a reference. Additionally, JGA guidelines [15] defined the input seismic motion in one specific direction, which was used in the seismic evaluation system. Therefore, in this study, the input was in one direction, and the 1/2 cross-section model (31,968 elements, 34,552 nodes) assumed symmetry. In addition, ground motion incoherence effects are not taken into account because the structure to be evaluated is small, and it is not considered in the design [15]. The hydraulic boundary at the ground surface was always set to zero excess pore water pressure such that the ground surface coincided with the groundwater level, and the lower end and sides were set to non-drainage conditions. Periodic boundaries were also set on each side.

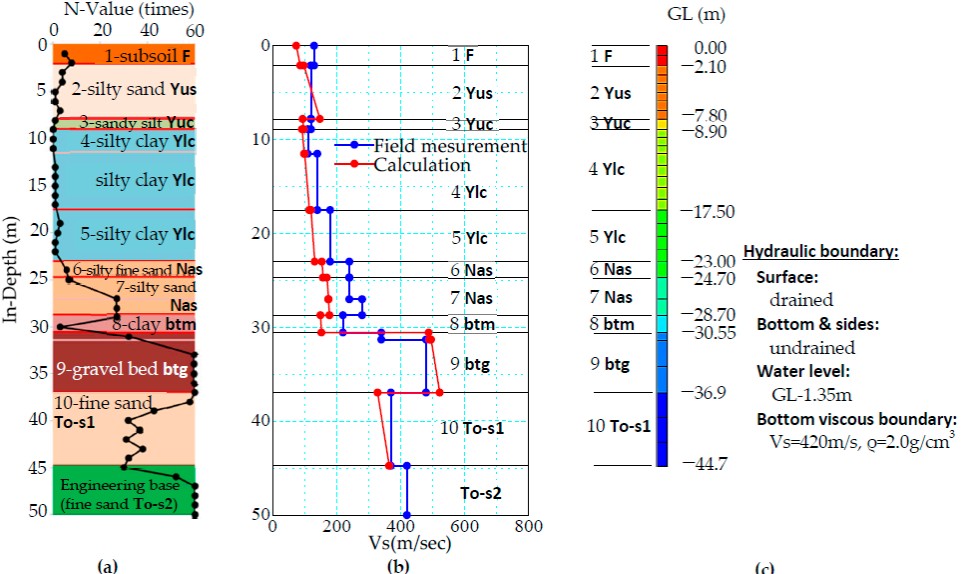

**Figure 1.** Soil boring log and 1D FE mesh with boundary conditions: (**a**) soil log; (**b**) shear wave velocity comparison; (**c**) FE mesh and BCs.

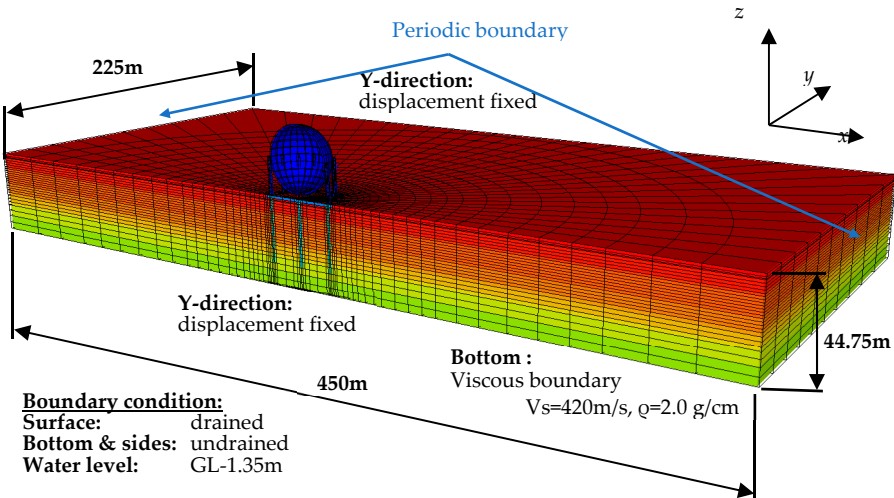

**Figure 2.** 3D FE mesh of ground and holder foundation.

Table 1 lists the elasto-plastic properties of the soils used in this analysis. The material constants and state quantities were determined using the SYS Cam-clay model [12] to reproduce various mechanical tests of undisturbed samples collected from the target points. This did not simply involve reproducing the mechanical test results but instead taking the state at the time of deposition as the initial state and numerically reproducing the series of processes from sampling, testing machine set, and then laboratory test. This allowed for the estimation of state quantities during deposition using inverse analysis through the reproduction of the mechanical test results [22]. Figures 3–5 show the standard consolidation test, undrained triaxial compression test, and deformation properties due to the repeated loading of the Ylc layer as an example of the reproduced results. All test results were reproduced with the same material constants and initial values. For the initial values of the ground, it was assumed that the specific volume (density), degree of structure, stress ratio, and degree of anisotropy were uniform within the same layer, and the overconsolidation ratio was distributed according to the overburden pressure [23,24]. When the shear wave velocity $V_s$ of each layer was calculated based on the elasto-plastic properties of the soil, the distribution of the initial values obtained by the calculation roughly agreed with the results of the field measurements, as shown in Figure 1b. In this elasto-plastic analysis, the damping factor used in the equivalent linear analysis [25] is not used, whereas the variation of equivalent shear stiffness $G_{eq}$ is specifically considered.

**Table 1.** Soil material constants used for analysis.

| | Refilled Soil Layer F | Yurakucho Formation Yus | Yurakucho Formation Yuc(1-2) | Yurakucho Formation Ylc(1-3) | Yurakucho Formation Ylc(1-4) | Nanagochi Formation Nas(1-5) | Nanagochi Formation Nas(1-6) | Buried Terrace Formation blm(1-7) | Buried Terrace Formation btg | Tokyo Formation To-s1(1-8) |
|---|---|---|---|---|---|---|---|---|---|---|
| **In-depth** G.L. (m) | −2.10 | −7.80 | −8.90 | −17.5 | −23.0 | −24.70 | −28.70 | −30.55 | −36.95 | −44.75 |
| **Elasto-plastic parameters** | | | | | | | | | | |
| Compression index $\tilde{\lambda}$ | 0.125 | 0.050 | 0.170 | 0.280 | 0.280 | 0.065 | 0.065 | 0.092 | 0.05 | 0.057 |
| Swelling index $\tilde{\kappa}$ | 0.005 | 0.003 | 0.012 | 0.019 | 0.018 | 0.0043 | 0.0038 | 0.0044 | 0.001 | 0.0017 |
| Critical state constant M | 1.30 | 1.60 | 1.60 | 1.40 | 1.70 | 1.43 | 1.43 | 1.40 | 1.20 | 1.45 |
| Specific volume at $q = 0$ and $p' = 98.1$ (kN/m$^2$) on NCL | 2.085 | 2.125 | 2.250 | 2.82 | 2.86 | 2.01 | 1.965 | 1.895 | 1.980 | 1.800 |
| Poisson's ratio $\nu$ | 0.1 | 0.2 | 0.1 | 0.1 | 0.1 | 0.25 | 0.25 | 0.1 | 0.1 | 0.13 |
| **Evolution rule parameters** | | | | | | | | | | |
| Degradation index of overconsolidation $m$ | 0.50 | 0.35 | 8.00 | 9.00 | 9.00 | 8.00 | 17.00 | 8.00 | 0.08 | 25.0 |
| Degradation index of structure $a$ ($b = c = 1$) | 5.00 | 10.0 | 1.00 | 0.65 | 1.0 | 4.00 | 8.00 | 0.40 | 2.20 | 4.00 |
| Ratio of $-D_y^p$ to $\left\|D_s^p\right\| C_s$ | 1.00 | 0.80 | 0.80 | 0.40 | 0.70 | 1.00 | 1.00 | 1.00 | 1.00 | 1.00 |
| Evolution index of rotational hardening $br$ | 3.50 | 0.10 | 0.20 | 0.20 | 0.20 | 3.00 | 3.00 | 0.10 | 3.50 | 3.00 |
| Limit of rotational hardening $m_b$ | 0.70 | 0.65 | 1.00 | 1.00 | 1.00 | 0.50 | 0.50 | 1.00 | 0.90 | 1.00 |
| **Physical properties** | | | | | | | | | | |
| Permeability $k$ (cm/s) | $2.58 \times 10^{-3}$ | $3.51 \times 10^{-4}$ | $1.29 \times 10^{-7}$ | $1.0 \times 10^{-7}$ | $1.0 \times 10^{-7}$ | $1.0 \times 10^{-6}$ | $5.79 \times 10^{-4}$ | $1.0 \times 10^{-6}$ | $8.25 \times 10^{-3}$ | $6.53 \times 10^{-3}$ |
| Density of soil particles $\rho_s$ (g/cm$^3$) | 2.030 | 2.735 | 2.765 | 2.625 | 2.626 | 2.684 | 2.747 | 2.672 | 2.000 | 2.663 |
| **Initial conditions** | | | | | | | | | | |
| Specific volume $v_0$ | 2.075 | 2.125 | 2.230 | 2.734 | 2.750 | 1.980 | 1.919 | 1.950 | 1.620 | 1.780 |
| Degree of structure $1/R_0^*$ | 1.60 | 1.10 | 1.60 | 1.40 | 1.90 | 1.20 | 1.10 | 5.20 | 1.00 | 3.00 |
| Overconsolidation ratio $1/R_0$ | 1.00 | 1.00 | 1.00 | 1.00 | 1.00 | 0.90 | 0.90 | 0.90 | 0.90 | 0.60 |
| Degree of anisotropy $\zeta_0$ | 1.00 | 1.00 | 1.00 | 1.00 | 1.00 | 0.90 | 0.90 | 0.90 | 0.90 | 0.60 |

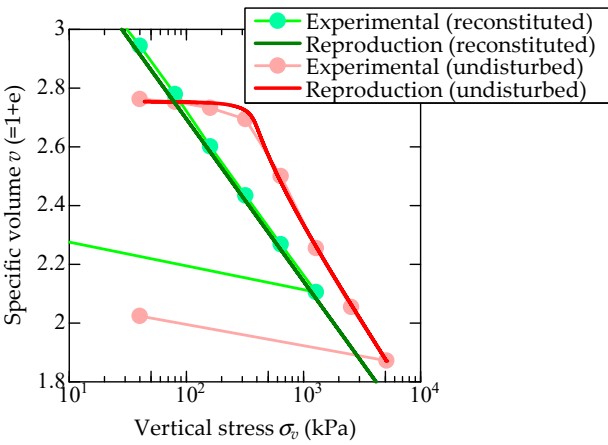

**Figure 3.** Standard consolidation test of the Ylc layer (experimental and reproduction results).

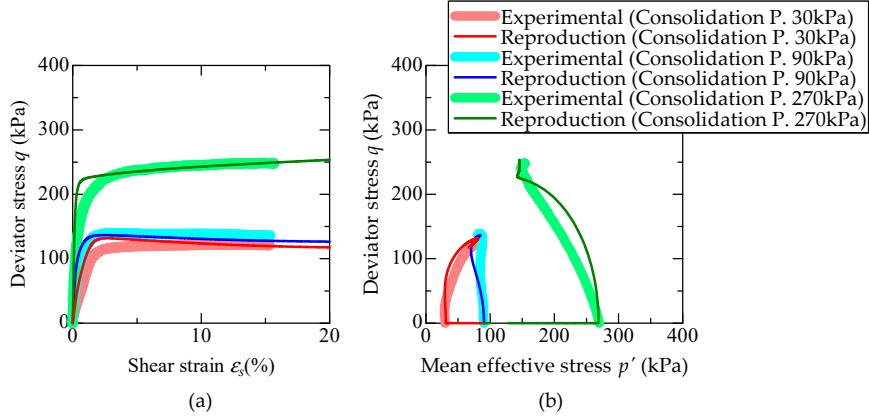

**Figure 4.** Undrained triaxial test of the Ylc layer (experimental and reproduction results): (**a**) Shear strain; (**b**) Mean effective stress.

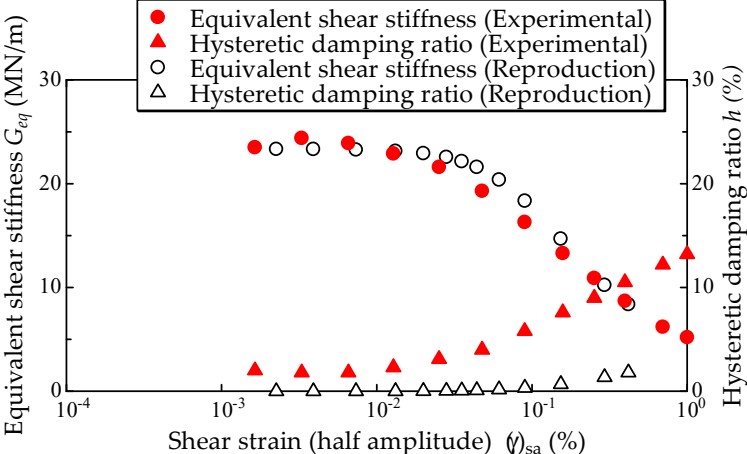

**Figure 5.** Dynamic deformation properties of the Ylc layer (experimental and reproduction results).

## 2.2. Holder Modeling

Figure 6 shows a schematic diagram of the holder to be analyzed, and Table 2 lists its specifications. An analysis model was created based on these specifications; Figure 7 shows the finite element mesh diagram of the holder, and Table 3 lists the material constants of the holder, foundation, and pile. All structural members were modeled using hexahedral solid elements with a single-phase hypoelastic material [26]. As shown in Figure 6, the

actual structure has four piles arranged per footing. Each footing was treated as a single unit in this analysis, and the four piles were modeled as a single solid element such that the flexural rigidity was equivalent to that of the actual product. The elastic modulus was set according to the actual material of the structural members other than the columns; however, the elastic modulus of the columns was set in consideration of the presence of the brace so that the natural period of the entire holder matched the actual value of 0.786 s (see Appendix A for the adjustment method). The piles were modeled as a bilinear elastic body by reducing the elastic modulus to 1/200 [27] when the relationship between the axial force N and bending moment M calculated from the solid elements exceeded the region of the M–N failure diagram [28] of the piles, as shown in Figure 8.

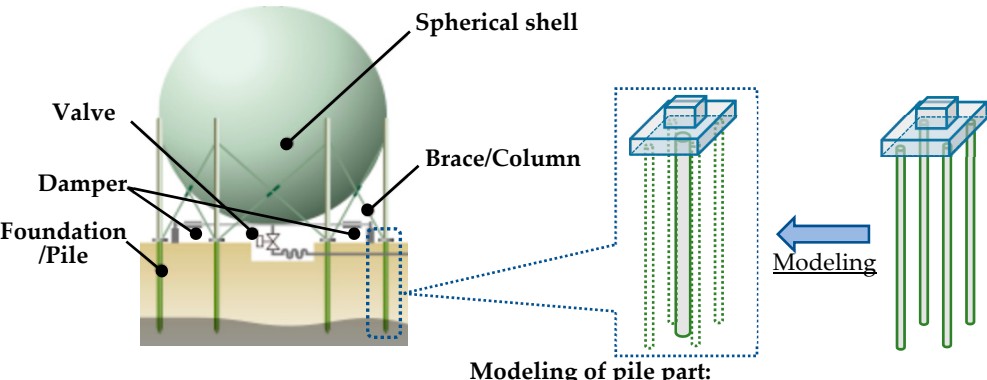

**Modeling of pile part:**
Each column is supported by four piles, which are replaced by a single stiffness-equivalent model in the analysis.

**Figure 6.** Spherical gas holder (analysis subject).

**Table 2.** Holder specifications.

| Items | Value |
|---|---|
| Nominal capacity | 200,000 m$^3$ |
| Design pressure | 0.83 MPa |
| Inner diameter of the spherical shell | 35.560 m |
| Inner diameter of the foundation | 35.330 m |
| Holder equatorial height | 19.000 m |
| Thickness of the spherical shell plate | 35.0 mm |
| Outer diameter, thickness, and number of column | φ600 mm × 8 mm × 14 pcs. |
| Material of the spherical shell and columns | High tensile steel (JIS G3128; WEL-TEN870C) |
| Outer diameter of the braces | φ90 mm |
| Material of the braces | High strength steel (HBS G3102; HT690) |

**Table 3.** Holder foundation/body material constants.

| | Unit Volume Weight (kN/m$^3$) | Elastic Modulus (kN/m$^3$) | Poisson's Ratio |
|---|---|---|---|
| Footing, Core | 25.00 | $2.35 \times 10^7$ | 0.2 |
| Radial beam | 18.86 | $1.07 \times 10^6$ | 0.2 |
| Underground beam | 22.66 | $1.85 \times 10^7$ | 0.2 |
| Perimeter reinforcement beam | 15.09 | $0.77 \times 10^7$ | 0.2 |
| Column | 0.448 | $1.70 \times 10^7$ | 0.3 |
| Spherical shell | 3.953 | $2.10 \times 10^8$ | 0.3 |
| Pile under the column | 2.935 | $3.19 \times 10^5$ | 0.2 |
| Pile under the core | 2.390 | $2.22 \times 10^5$ | 0.2 |

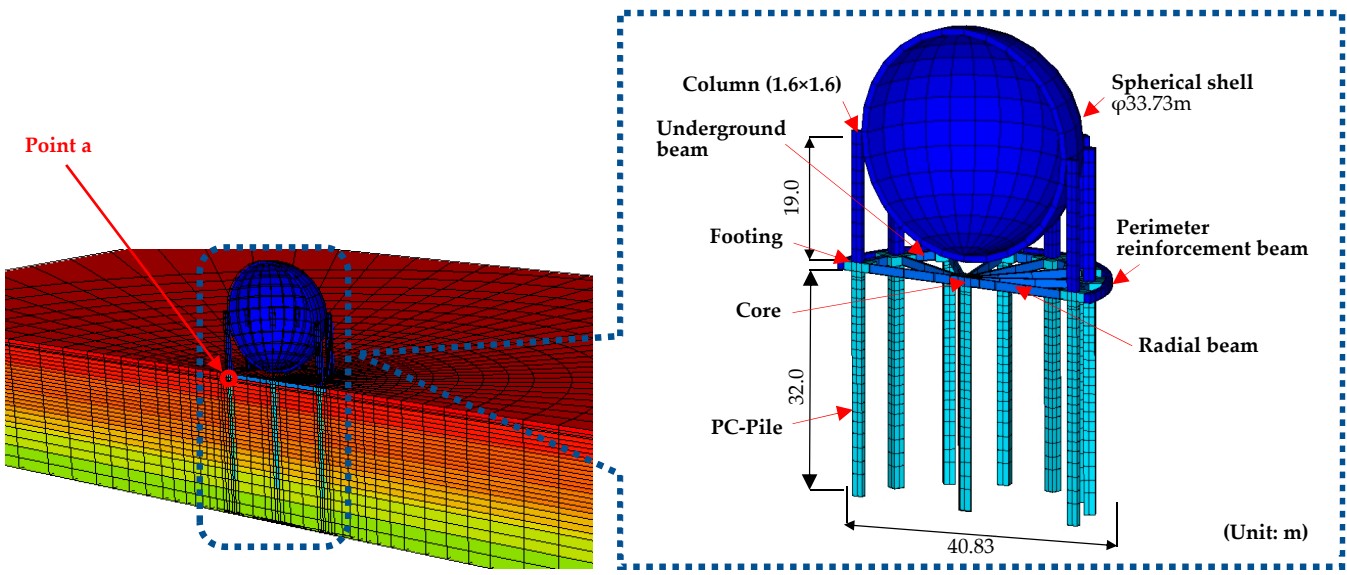

**Figure 7.** Analysis model.

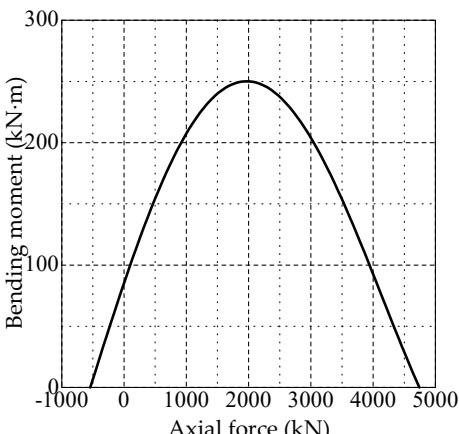

**Figure 8.** Pile M–N failure diagram.

The upper load acting on the ground (dead weight of the holder and pile) was shared by the bottom foundation (radiant and annular beams in this analysis), the tip-bearing force (axial force of the pile), and the frictional force on the peripheral surface of the pile. Assuming that the pile is solely responsible for the entire holder load of approximately 6800 kN, the axial force acting on each pile is approximately 240 kN. This is described in more detail later, but the axial force acting on a single pile at the time prior to the earthquake input is approximately one-half; thus, the difference is shared by the bottom foundation and the peripheral surface of the pile.

## 3. Input Seismic Motion

In this study, the effects of the duration of seismic motion were evaluated by selecting two types of seismic records from recent damaging earthquakes and using an input seismic motion whose amplitude was adjusted to obtain the same acceleration response spectrum (for L2 seismic motion settings). Figure 9 shows the two input seismic motions. Seismic motions (A) and (B) were set as the strong ground motion waveforms of the "NS-direction seismic record at the Muroran Port during the 1993 southwest-off Hokkaido earthquake" [29] and the "NS-direction seismic record at the Hanshin Expressway East-Kobe Bridge (GL-35 m) during the Hyogo-ken Nanbu earthquake" [30], respectively, whose amplitudes were adjusted with the spectrum used to set the JGA L2 seismic motion

(Figure 10). As shown in Figure 10, the original waveforms of both seismic motions differ significantly in magnitude. The original waveform used for seismic motion (A) was small in magnitude; thus, the amplitude was significantly adjusted in the wide period band from the short -to long-period components. However, the waveform used for seismic motion (B) had minor adjustments because it was an L2 earthquake close to the design spectrum. Seismic motion (B) was the seismic wave used in a previous study [13]. Both seismic motions were strong ground motions that included long-period components. In each case, the amplitude was adjusted without changing the duration. The duration was defined by the "bracketed duration" of Bolt [31] and the duration based on the power accumulation time of Trifunac and Brady [32]. The "bracketed duration" is defined as the length of time between the first and the last time when the amplitude of the seismic record exceeds a "given level". This level was defined as "10% of the maximum amplitude", as in Kamiyama [33] (hereinafter referred to as "b-duration $t_b$"). The definition by Trifunac and Brady is to calculate the duration as the length of the interval containing 90% (5~95%) of the total power of the earthquake record (hereafter referred to as "p-duration $t_p$"). The duration $t_b$ and $t_p$ calculated from these definitions were shown in Table 4 for both adjusted and original waves. The table shows that the duration of earthquake motion (A) is about 9 times longer than that of earthquake motion (B) in terms of $t_b$.

The seismic response analysis used a bottom viscous boundary [22,34,35] corresponding to shear wave velocity Vs = 420 m/s and density ρ = 2.0 g/cm³ in the horizontal direction of all the nodes on the ground bottom, and the seismic waves were input equally in the horizontal direction of all the nodes on the ground bottom.

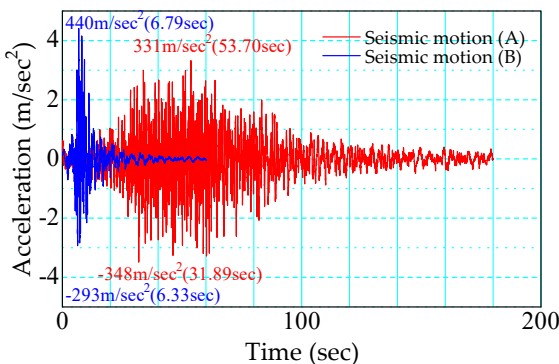

**Figure 9.** Input seismic motion for analysis: seismic motions (A) and (B).

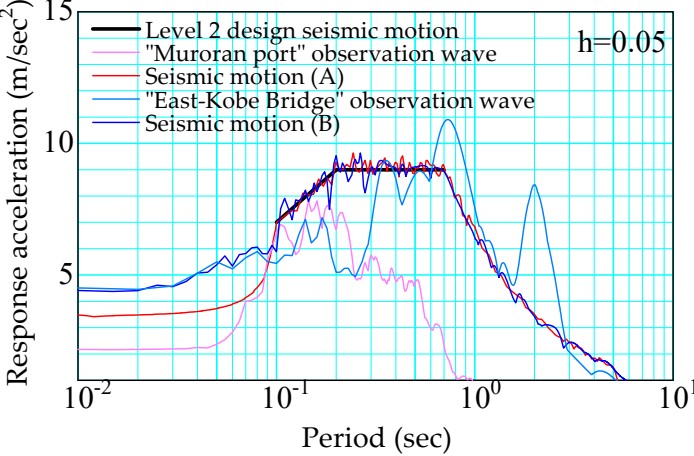

**Figure 10.** Acceleration response spectrum.

**Table 4.** Duration of seismic motion (A) and (B). The upper part of each item in the table shows the value of the adjusted wave, and the lower part (in parentheses) shows the value of the original seismic wave.

| | Maximum Acc. (gal) | b-Duration $t_b$ (s) | p-Duration $t_p$ (s) |
|---|---|---|---|
| | | Start-End Time of $t_b$ (s) | Start-End Time of $t_p$(s) |
| Seismic motion (A) | 348 (217) | 171.4 (72.8) 0.0–171.4 (16.2–89.0) | 78.4 (45.2) 25.9–104.3 (31.0–76.2) |
| Seismic motion (B) | 440 (443) | 19.5 (13.3) 2.1–21.6 (5.3–18.7) | 11.7 (8.4) 5.3–17.0 (6.1–14.5) |

In this study, the seismic response of the ground was specifically discussed. Therefore, 1D analysis was conducted to examine the response of the ground itself when two different seismic motions were input to the target ground. Figures 11 and 12 show the settlement amount during and after the earthquake when seismic motions (A) and (B), respectively, were input to the horizontal ground using the FE mesh and boundary conditions in Figure 1c and when the structure was not installed for each (i.e., ground surface settlement amount and stratified settlement amount). The ground surface settlement amount was 0.44 m and 0.21 m for seismic motions (A) and (B), respectively. The mean effective stress of the surface sandy soil layer (B layer/Yus layer) decreased during the earthquake (i.e., excess pore water pressure accumulated); consolidation settlement began as a result of the dissipation of the excess pore water pressure during the earthquake for the B layer and immediately after the earthquake for the Yus layer. Figures 13 and 14 show the horizontal displacements of seismic motions (A) and (B), respectively, on the ground surface. Comparing the seismic motion effects, it can be observed that the horizontal displacement during the earthquake was larger for seismic motion (A), which has a longer duration that significantly disturbed the ground.

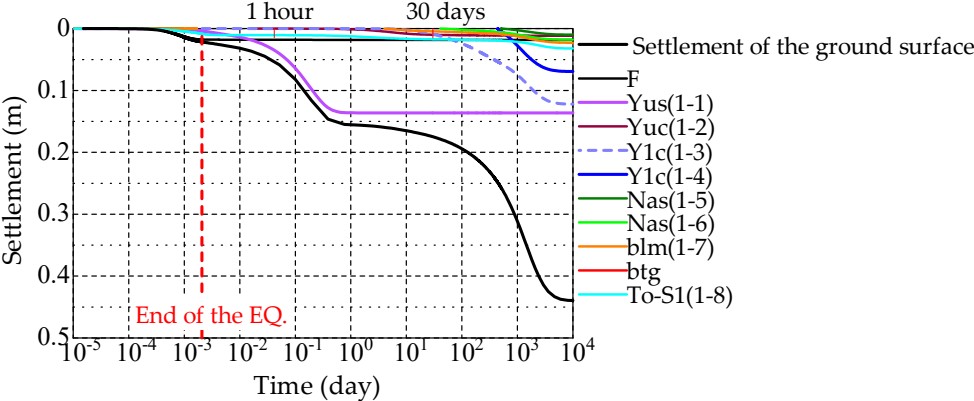

**Figure 11.** Settlement behavior during and after earthquake: seismic motion (A).

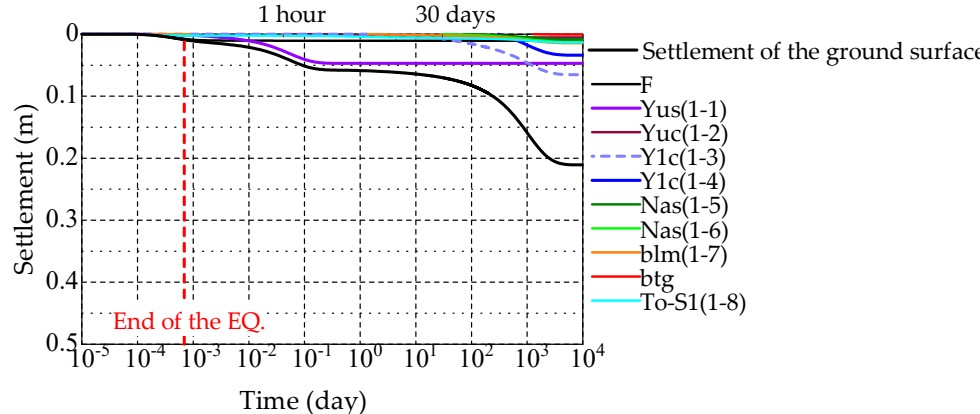

**Figure 12.** Settlement behavior during and after earthquake: seismic motion (B).

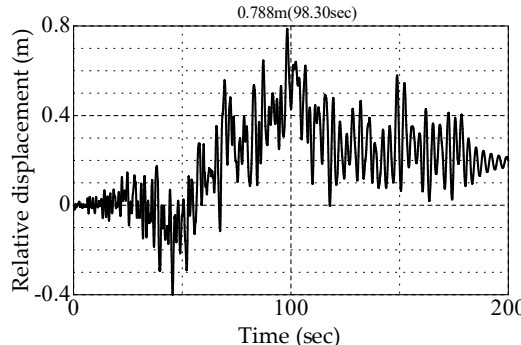

**Figure 13.** Horizontal displacement in ground surface: seismic motion (A).

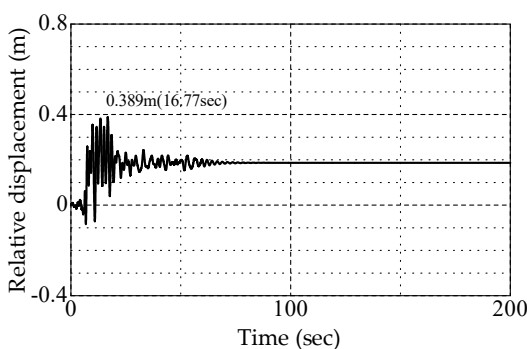

**Figure 14.** Horizontal displacement in ground surface: seismic motion (B).

## 4. Behavior of Coupled Holder–Pile–Ground System during Earthquake

### 4.1. Seismic Performance Evaluation of Gas Holder during Long-Term Strong Ground Motion

For the long-duration seismic motion (A), Figure 15 shows the change in the mean effective stress over time, Figure 16 shows the settlement behavior during and after an earthquake at point a in Figure 7, and Figure 17 shows the progress of the pile damage. The mean effective stress was almost zero in the surface sandy soil layer (B layer/Yus layer) at approximately 10 s after the start of the earthquake, and liquefaction occurred (along with the occurrence of positive excess pore water pressure). The duration of the seismic motion was long; thus, the mean effective stress of the deep soft clay layer also gradually decreased. The holder showed approximately 0.1 m of settlement during the earthquake, as well as 0.37 m of settlement after the earthquake due to consolidation associated with the dissipation of positive excess pore water pressure that accumulated during the earthquake. However, this did not result in any inclination associated with unequal settlement, which would hinder post-earthquake facility service (i.e., continuous use). Focusing on the degree

of damage to the pile, "damage" (i.e., reaching the failure line of the M–N diagram) began at the pile head approximately 12 s after the earthquake occurrence, and the pile damage gradually progressed. With the exception of parts shallower than GL −15 m, the pile was damaged in almost all sections after 70 s in the latter half of the main motion.

Next, in Figure 16, we show the results of the seismic response analysis implemented with the without-piles model along with the settlement behavior during and after the earthquake to confirm the extent to which the presence of the pile contributed to reduced seismic damage (i.e., settlement suppression). The amount of settlement of the holder foundation was 0.4 m during the earthquake and ultimately approximately 0.7 m, which was roughly double that of the with-piles model. In other words, these results indicate that although the pile was damaged in almost all sections during seismic motion (A), the presence of the pile was effective in suppressing the holder settlement. The settlement suppression mechanism is discussed in the following section.

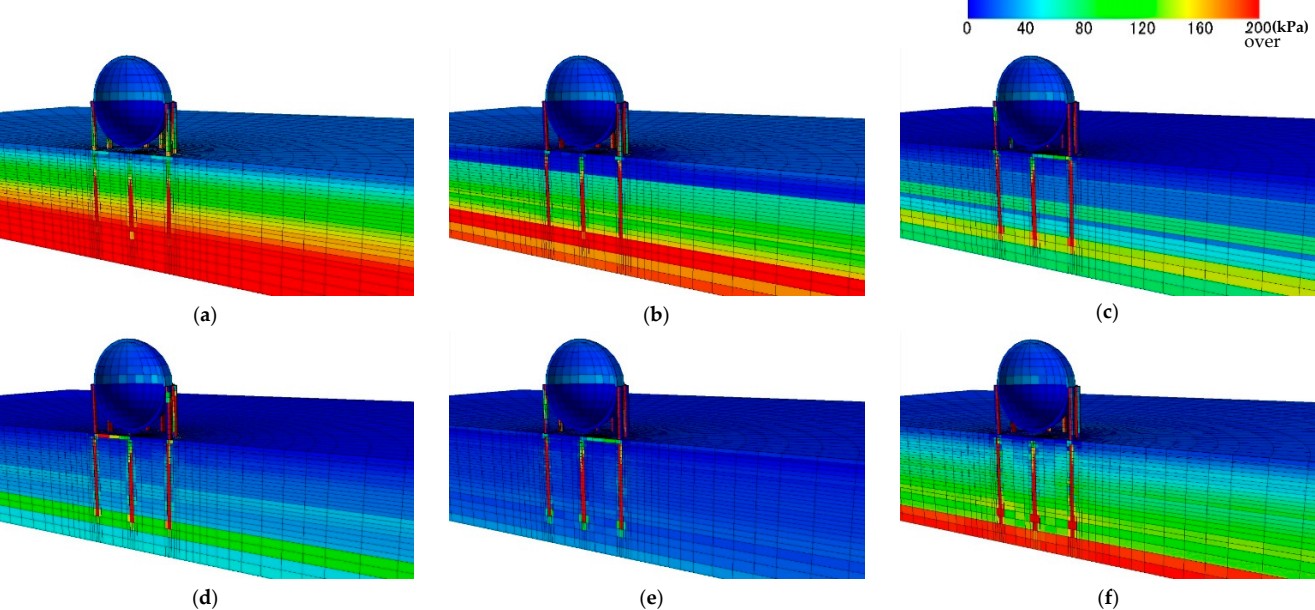

**Figure 15.** Change in mean effective stress over time, seismic motion (A): (**a**) immediately before the EQ; (**b**) 12 s after the start of the EQ; (**c**) 30 s; (**d**) 38.5 s; (**e**) end of the EQ; (**f**) end of consolidation (30 years after the EQ).

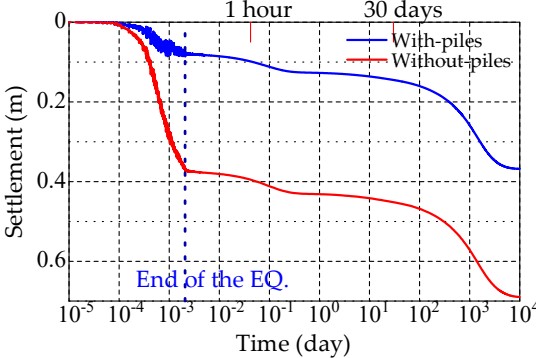

**Figure 16.** Comparison of settlement behavior during and after earthquake between with and without piles: seismic motion (A).

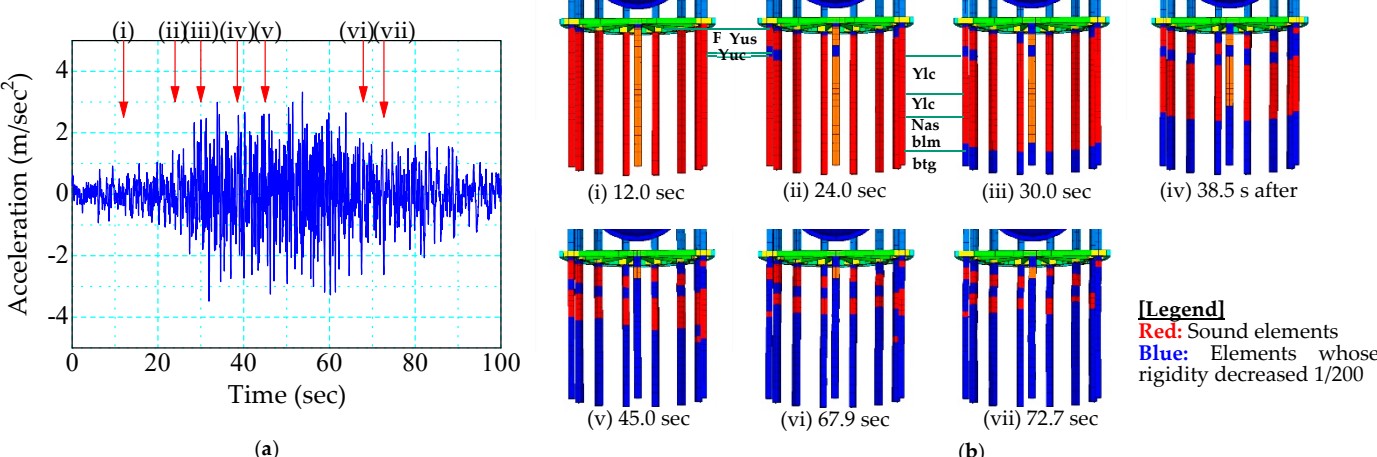

**Figure 17.** Status of pile damage progress: seismic motion (A): (**a**) input seismic motion and damage time; (**b**) pile damage.

*4.2. Settlement Suppression Mechanism by Pile (Pile Member Force Evaluation)*

Figure 18 shows the changes in the axial force distribution acting on a typical pile at point a in Figure 7. The axial force acting on the pile increased from during the earthquake to afterwards. The changes in the axial force of the pile due to seismic motion can be interpreted as follows:

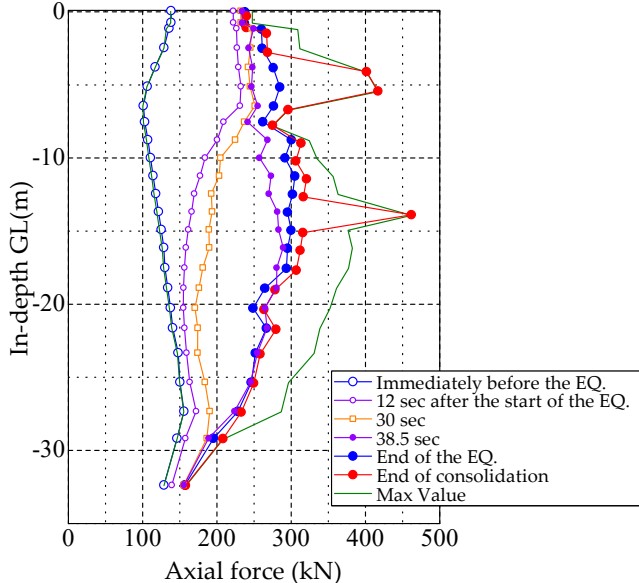

**Figure 18.** Changes in axial force distribution of pile over time: seismic motion (A).

(1) Prior to earthquake occurrence

The axial force acting on the pile head was approximately 120 kN and was almost constant in the depth direction. As described in Section 2.2, in addition to this axial force, the holder load is shared by the bottom foundation and the frictional force between the peripheral surface of the pile.

(2) From soon after to 12 s after the start of the earthquake

As shown in the one-dimensional analysis (Figure 12), the B layer began to undergo consolidation settlement immediately after the earthquake. Because the pile was present, the force that supports the bottom foundation (ground reaction force) was lost owing to the settlement of the B layer, and the axial force of the pile increased to approximately 220 kN at the pile head. Additionally, the axial force of the pile did not increase significantly at this stage in the deep clay layer.

(3) From 12 s after the start of the earthquake to the end

The repeated loads caused by the seismic motion resulted in a decreased mean effective stress (i.e., effective confining pressure) in the sandy soil layer. This decrease in the effective confining pressure results in a positive excess pore water pressure and an increased axial force on the pile. It is also generally inferred that a clay layer is less susceptible to seismic damage. However, the Ylc layer was not only in an extremely soft state, but the duration of the seismic motion (A) was long, and a large shear deformation occurred in the ground, as shown in Figure 12. Therefore, the clay layer experienced further decreases in the effective confining pressure and alternatively gradual increases in the axial force of the pile, even in the deep part.

(4) During consolidation settlement (after the earthquake)

After the end of the earthquake, consolidation settlement occurred first in the liquefied sand layer and then in the clay layer after a delay. This consolidation settlement caused negative friction, and the axial force greatly increased in locations where the piles were undamaged.

In this way, the consolidation settlement in the B layer during the earthquake in (2) and the decreased effective confining pressure of the sandy Yus and clay Ylc layers during the earthquake in (3) caused stress redistribution, increasing the pore water pressure and axial force of the pile. From the M–N failure diagram in Figure 8, it can be observed that the increased axial force of the pile increased the allowable bending moment (i.e., increased the yield strength of the pile). Figure 19 shows the shifts in the M–N failure diagram of the member force acting on the pile during the earthquake as well as its bending moment diagram; indeed, the bending moment acting on the pile during the earthquake increased, causing gradual damage to the pile and maintaining its support functions for some period of time. Therefore, although the pile was ultimately damaged in almost all sections, the deformation of the holder ground was suppressed to some extent. Figure 20 shows the shear strain distribution in the without-piles and with-piles models. The shear strain was the maximum value until the end of the earthquake. The without-piles model shows a concentration in the region where the shear strain is predominant near the surface layer, but the with-piles model shows that the shear strain was predominant around the pile as well as near the surface layer, and it can be observed that the pile resisted the shear deformation by the seismic motion.

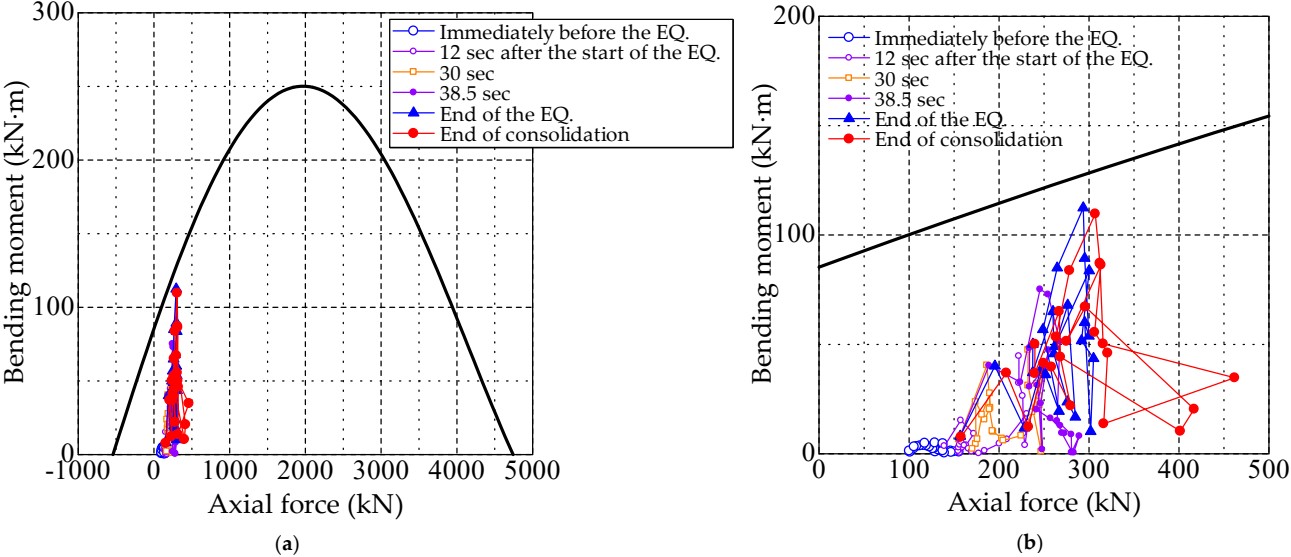

**Figure 19.** Plot on M–N failure diagram and changes in the bending moment over time, seismic motion (A): (**a**) M–N failure diagram; (**b**) enlarged view of (**a**).



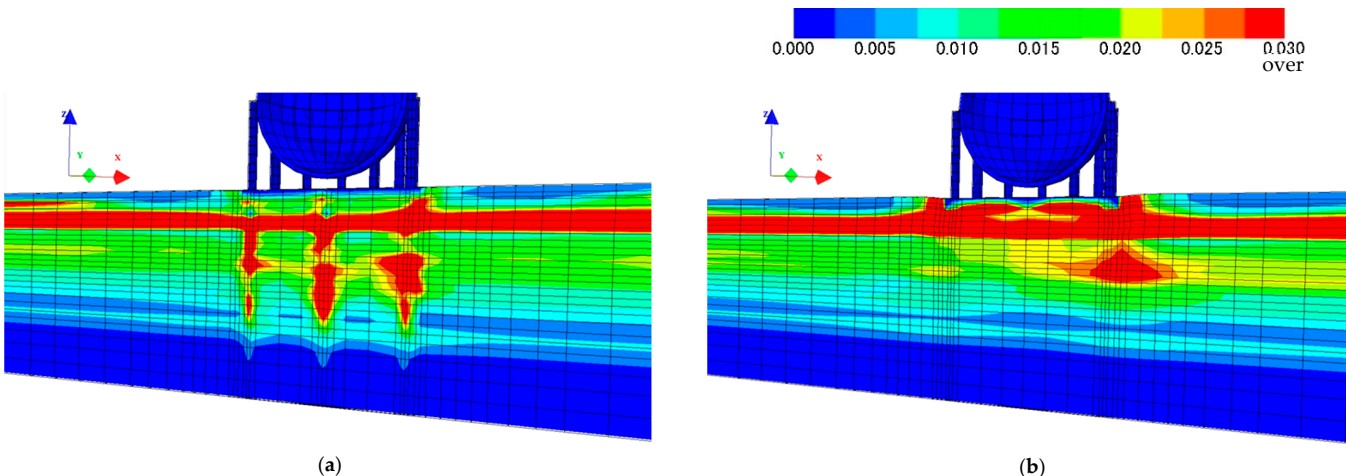

(**a**)  (**b**)

**Figure 20.** Shear strain distribution (maximum value until the end of an earthquake) of (**a**) with-piles and (**b**) without-piles models during seismic motion (A).

Okada et al. [36] and Hoshikuma [37] used pre-built RC pile foundation models to conduct alternating loading experiments in which displacements up to 20 times that of the yield displacement of the piles were gradually input. The results showed that the pile could roughly support a predetermined vertical load even in cases where the shear failure occurred in some of the piles owing to progressive damage of the piles. In other words, it can be inferred from the analysis results in this study that the pile damage gradually progressed, and the pile suppressed settlement until approximately 140 s (when the pile was completely damaged), similar to the experimental results.

### 4.3. Effect of Duration on Pile Support

Next, the results of the short-duration seismic motion (B) analysis conducted in the previous study [13] were republished and compared with seismic motion (A) to evaluate how differences in the input seismic motion affect the amount of displacement of the holder ground and the pile support function. Figure 21 shows the changes in the mean effective stress over time, Figure 22 shows the settlement behavior during and after the earthquake at point a in Figure 7, and Figure 23 shows the progress of the pile damage. The holder foundation settled approximately 0.1 m during the earthquake and ultimately 0.3 m after the earthquake owing to long-term consolidation settlement. However, the settlement amount was small compared to that from the seismic motion (A). The mean effective stress was almost zero in the surface sandy soil layer (B layer/Yus layer) approximately 5 s after the occurrence of the earthquake, resulting in liquefaction. The mean effective stress of the deep clay soil layer also slightly decreased with time, but the seismic motion duration was short; thus, this decrease was not as prominent as that for seismic motion (A). Focusing on the extent of pile damage, almost all areas were damaged in the five-second period from 4.0–9.0 s after the earthquake, and unlike that with seismic motion (A), damage occurred within a very short period of time. Figure 22 also shows the settlement behavior during and after the earthquake in the without-piles model. The settlement amount of the holder foundation was approximately 0.1 m during the earthquake and ultimately approximately 0.3 m, which was almost equal to that of the with-piles model. In other words, unlike the case of seismic motion (A), the settlement suppression effect due to pile presence could not be observed.

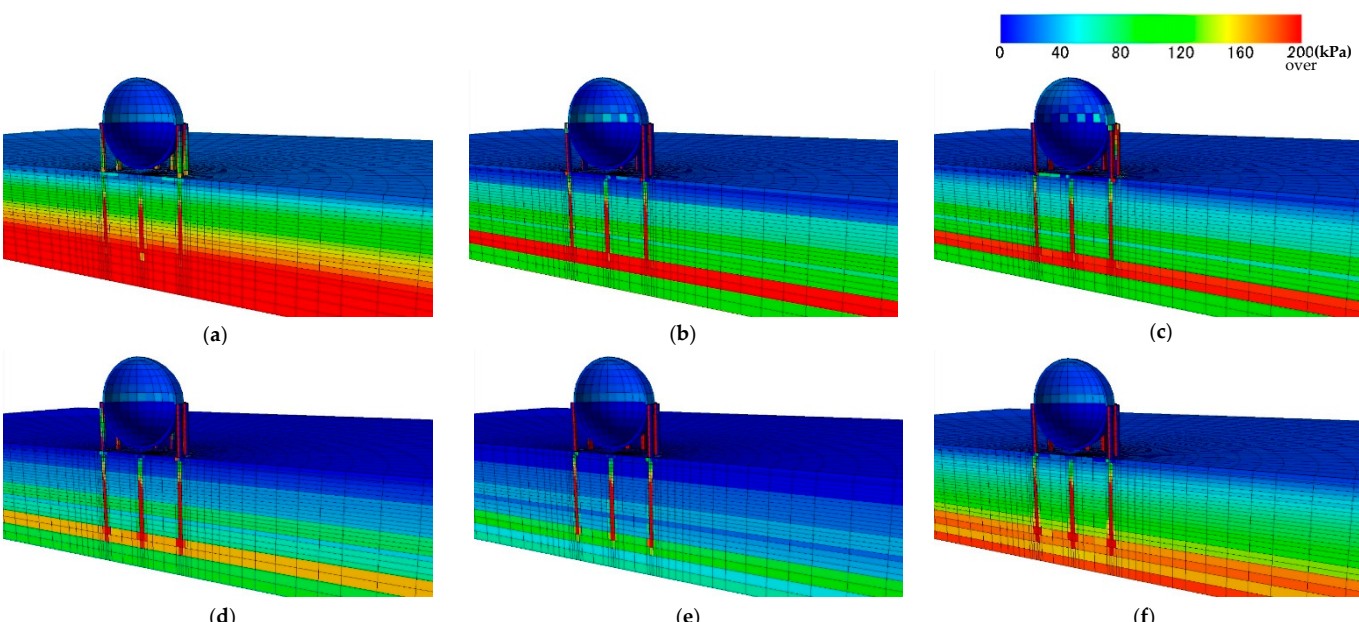

**Figure 21.** Changes in mean effective stress over time, seismic motion (B): (**a**) immediately before the EQ; (**b**) 7 s after the start of the EQ; (**c**) 7.5 s; (**d**) 10.5 s; (**e**) end of the EQ; (**f**) end of consolidation.

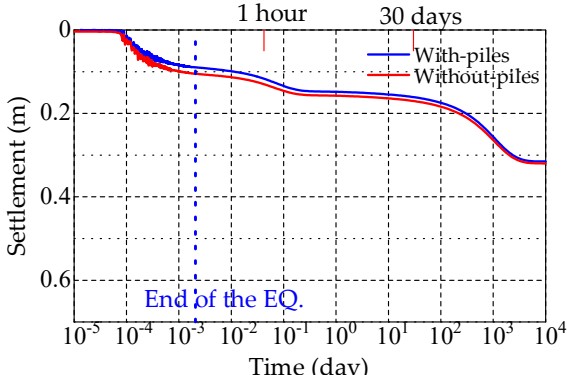

**Figure 22.** Comparison of settlement behavior during and after earthquake between with and without piles: seismic motion (B).

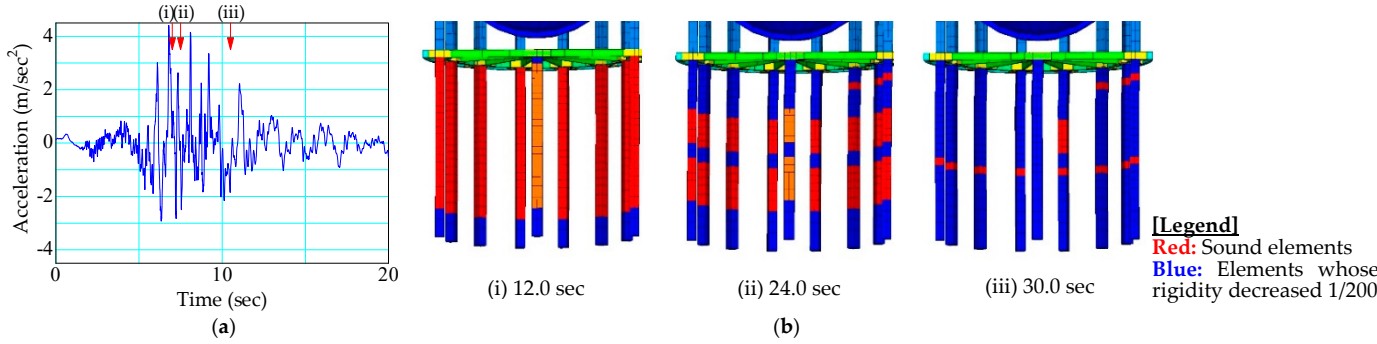

**Figure 23.** Progress of pile damage, seismic motion (B): (**a**) input seismic motion and damage time; (**b**) pile damage.

The axial force distribution acting on the pile at point *a* in Figure 7 is shown in Figure 24, which shows that the settlement suppression effect of the pile was not observed for seismic motion (B). The case of seismic motion (B), in which a large maximum acceleration occurred in a very short time period, was similar in that the force supported by

the bottom foundation (ground reaction force) was lost immediately after the earthquake occurrence owing to the settlement of the B layer, and the axial force of the pile increased. However, unlike the case of seismic motion (A), no progressive increases in the axial force were observed during the earthquake, and all parts of the pile were damaged. In other words, the support function of the pile was immediately lost without any sign of increased resistance of the pile due to decreased ground rigidity, as was the case in seismic motion (A); therefore, the settlement suppression effect could not be obtained. Figure 25 shows the shear strain distribution of the with-piles and without-piles models. Unlike seismic motion (A), there were no differences between the presence or absence of pile modeling, and the shear strain was predominant only near the ground surface. It can be observed from this as well that the piles were damaged without resisting seismic motion and that no support functions were observed.

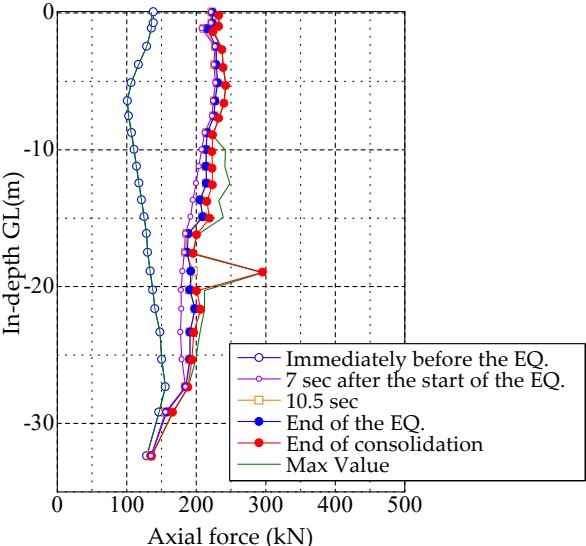

**Figure 24.** Changes in the axial force distribution of pile over time: seismic motion (B).

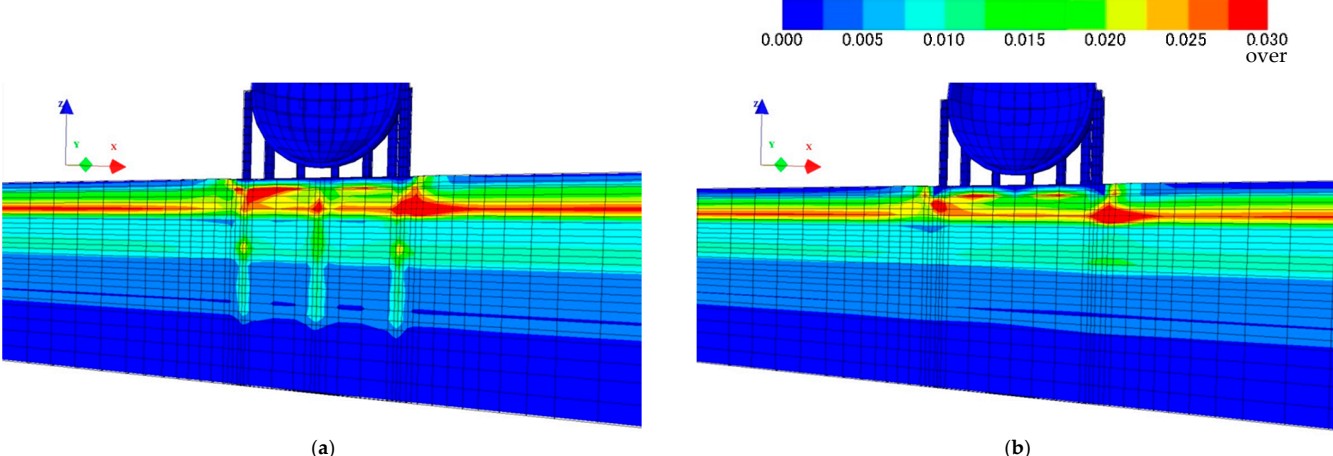

(**a**) (**b**)

**Figure 25.** Shear strain distribution (maximum value until the end of an earthquake) of (**a**) with-piles and (**b**) without-piles models: seismic motion (B).

## 5. Conclusions

We focused on an actual pile-supported spherical gas holder on alternating layers of soft sand and clay by conducting a three-dimensional seismic response analysis on an interacting holder–pile–ground system and evaluated the seismic resistance of the gas holder in the event of strong ground motion that would damage the pile during an

earthquake. Here, we determined the effect that strong ground motions with the same acceleration response spectra but significantly different durations would have on the support function of the pile. When the acceleration response spectrum is identical, the maximum acceleration increases as the duration of the main motion decreases. The main conclusions of this study are as follows:

- With regard to the characteristics of the mechanical response of the target alternating ground, when the duration of the input seismic motions was different despite the response spectrum being identical, the long-duration seismic motion (seismic motion (A)) had larger ground vibrations relative to the bottom of the ground (horizontal displacement) (i.e., larger shear displacement) and greater disturbance in the ground relative to the short-duration seismic motion (seismic motion (B)). As a result, the ground settlement amount in the former was larger than that in the latter.

- The long-duration seismic motion (seismic motion (A)) exhibited a large settlement suppression effect due to the pile. This was because the axial force of the pile, as well as the positive pore water pressure, increased owing to the stress redistribution caused by the decrease in the effective confining pressure of the sand and clay layers during the earthquake, which increased the pile resistance (i.e., allowable bending moment). Consequently, when the pile damage gradually progressed with seismic motion (A), the pile could maintain support functions for a certain period of time, even if the pile was ultimately damaged in all sections.

- Furthermore, the short-duration seismic motion (seismic motion (B)) exhibited almost no settlement suppression effect due to the pile. In the case of seismic motion (B), an extremely high maximum acceleration acted on the ground in a very short period of time, and the piles were suddenly damaged in almost all sections. This was because the support function of the pile was lost almost instantaneously without the increased resistance of the pile, which was generated by the decreased ground rigidity, having had no time to appear, in contrast to the case of seismic motion (A).

- These results show that the ground response and pile failure modes can differ if the duration differs, even if the input ground motion has the same acceleration response spectrum. The results suggest that duration will need to be considered in the future when selecting seismic motions to be used in evaluations.

- In this study, we were able to verify the possibility of the collapse of the holder during strong L2 class ground motions, including those with long durations. As a result, we will take all possible safety measures to prevent the holders from malfunctioning even after a large-scale natural disaster occurs. In order to further improve this research, it is necessary to evaluate the effects of geological composition (such as different soil conditions, geological irregularities, and so on) and the input seismic motion (in the case of seismic response spectra containing more short-period components, and the influence of consecutive earthquakes). Therefore, we will increase the number of analysis cases to deal with these issues in the future.

**Author Contributions:** Conceptualization, M.K.; methodology, M.K. and T.N.; software, T.T.; validation, T.N. and K.N.; formal analysis, T.T.; investigation, M.K. and T.N.; resources, M.K.; data curation, M.K. and T.T.; writing—original draft preparation, M.K. and K.N.; writing—review and editing, M.K., K.N. and T.N.; visualization, M.K. and T.T.; supervision, A.A.; project administration, M.K.; funding acquisition, M.K. All authors have read and agreed to the published version of the manuscript.

**Funding:** This research received no external funding.

**Institutional Review Board Statement:** Not applicable.

**Informed Consent Statement:** Not applicable.

**Data Availability Statement:** No new data were created or analyzed in this study. Date sharing is not applicable to this study.

**Acknowledgments:** The authors are grateful to Naoto Ohbo of the Association for the Development of Earthquake Prediction. The authors are also grateful to Noritake Oguchi, Tokyo Gas Co., Ltd.

**Conflicts of Interest:** The authors declare no conflict of interest.

## Appendix A

Figure A1 shows a model diagram of the gas holder. The unit weight volume of the spherical shell was set such that the weight of the gas holder's body would be equal to the constant reaction force per column. Furthermore, the overall rigidity was determined by setting the elastic modulus of the column as follows: A total of 40 cycles of sine waves with various periods of 300 gal were applied in the *x*-axis direction of the lower end of the column of the model shown in Figure A1. The elastic modulus of the column was then set as the elastic modulus when the period at which the maximum horizontal displacement of the top was equal to the natural period of the gas holder (0.786 s) [13].

Figure A2 shows the relationship between the maximum horizontal displacement and elastic modulus of the spherical shell top when the input period was set as 0.786 s and when the column of the elastic modulus was changed. Figure A3 shows the relationship between the maximum horizontal displacement and period of the spherical shell top when the column elastic modulus was set to $1.7 \times 10^7$ kN/m$^2$, and a waveform with a changing period was the input. The value of $1.7 \times 10^7$ kN/m$^2$, which was the maximum horizontal displacement, was set as the elastic modulus of the column [13]. The material constants of the gas holder set based on the above are listed in Table 3 of the main text.

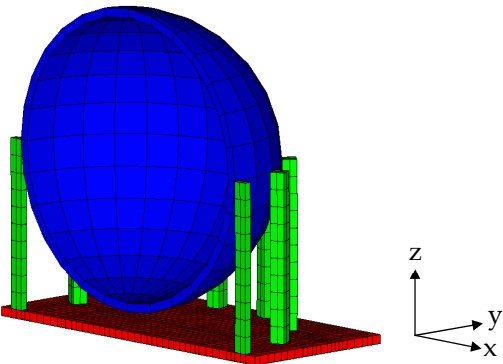

**Figure A1.** Gas holder modeling.

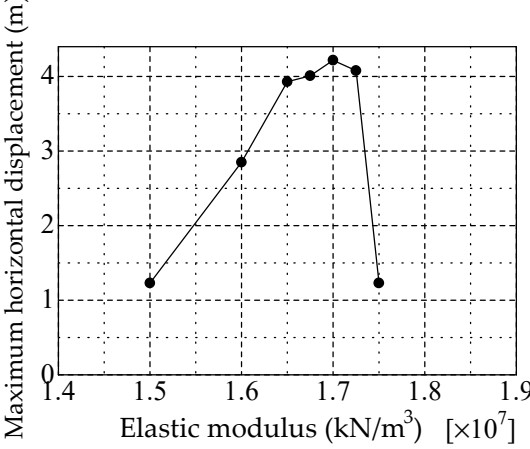

**Figure A2.** Elastic modulus and maximum horizontal displacement.

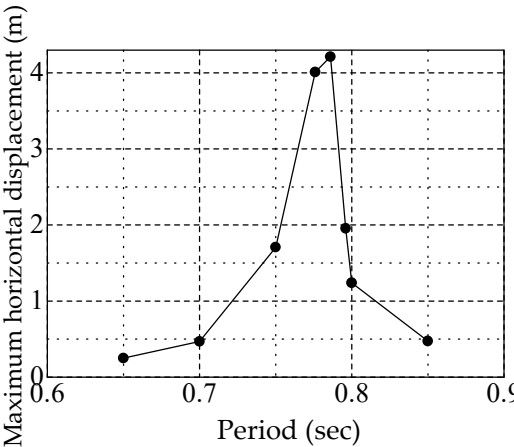

**Figure A3.** Each period and maximum horizontal displacement.

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
