# Peer review of "Effects of Strong Ground Motion with Identical Response Spectra and Different Duration on Pile Support Mechanism and Seismic Resistance of Spherical Gas Holders on Soft Ground"

_applsci, doi:10.3390/app112311152_

Round 1
Reviewer 1 Report
Major revision is suggested for this paper. The authors have to take care of the following points in the revised version of the paper:
1) page 1, line 34: define in terms of PGA, magnitude or else the 'large-scale natural disaster'.
2) page 2, line 58: I am not sure if 'turbulence' is the correct term.
3) page 2, lines 69-70: strong motions having a long duration are associated with a high (moment) magnitude, subduction zones, long period content and, usually, are recorded in soft soils. Please specify more precisely the source of long duration motions and which of those are you using herein.
4) page 2, line 77: add the following works as references: https://doi.org/10.1007/s10950-007-9066-y
https://doi.org/10.1115/PVP2004-3076
5) page 2, lines 78-81: By definition spectrum does not include duration. It is difficult to realize this adjustment in amplitudes for long-duration motions. Please explain. How do you define duration?
6) page 3, line 101: 'diluvium' is not the correct term.
7) page 3, figure 1 and corresponding text: values for shear modulus, shear wave velocity and damping ratios are missing. The authors make use of 1-D site response analysis. 2-D could be used as well and the results would more reliable. Please explain.
8) section 3, lines 178-181: same to comment 5).
9) section 3: 3-D analyses are executed but the authors do not provide evidence regarding the horizontal and vertical ground motion components used and they give the impression of using just one component. Please explain.
10) section 3: ground motion incoherence effects seem to have been ignored. Why?
Reviewer 2 Report
This is an interesting study and the authors have introduced an important and innovative topic tackling the assessment of seismic performance for spherical gas holders on soft ground under the effect of strong ground motion with identical response spectra. A three‐dimensional soil–water coupled finite model is conducted to study the seismic performance of spherical gas holders. The main focus in the study was on assessing the piles damage and soil interaction to strong ground motions. I think that this unique study and datasets has not been utilized to its full extent and the literature review should include more details, which are listed in the comments below. I recommend this paper to be published after considering the following minor comments:
- In the introduction a small paragraph should be added to discuss the past case studies regarding the damage of spherical gas holders and their main types, by which, the mechanical performance of these tanks should be introduced and how each criteria can affect the results such as the soil interaction and the type of supports and other criteria.
- It is better to add a paragraph in the introduction that discusses how this research is coping and helping in the implementation of the main sustainable development goals (SDG) that are employed by the United Nations (UN) Agenda. These, study is shedding the light on an important infrastructure, so the authors should add a paragraph discussing the importance of this system and how it can affect the social and economic prosperity of the societies, by giving some supportive case studies of past incidents. See the following reference:
- Li, Y. and Xi, X., 2017. Earthquake damage and countermeasure of industrial lifeline and equipment. In Earthquake Engineering Frontiers in the New Millennium(pp. 250-253). Routledge.
- Figure 1 can be founded in other papers, so it should be cited and referred to this paper:
- Kobayashi, M. and Takaine, T., 2018. Earthquake resistance evaluation of a spherical gas holder considering its ultimate state due to liquefaction-induced differential settlement of sandy ground. In of the 16th Asian Regional Conference on Soil Mechanics and Geotechnical Engineering.
- The models are built and simulated in appropriate and detailed way, however the seismic motions considered in this study are two, which I think should be at least five main ground motions to illustrate the results with more extensive details. Moreover, I suggest adding a comparative table or paragraph discussing and prioritizing the importance of each parameter in this study and comparing the results of this study with the results of analysis for short‐duration strong ground motion.
- A paragraph should be added in the conclusion to discuss how this research can help in future work and what are the main gaps that should be investigated.
Round 2
Reviewer 1 Report
Major revision is required for the following items:
1) Original comment 4: the 2 references suggested and added by the authors are completely irrelevant with the subject as they refer to buildings and not tanks. The authors have to cite the two references given in my original comment because they deal with tanks (no matter if they are liquid tanks) subjected to long duration motions.
2) Original comment 5: the authors do not provide any explanation about how they define duration. Please clarify how you define duration, using one or more duration indices (e.g. significant duration) found in literature, in order to apply the adjustment procedure.
3) Original comment 7: no sufficient explanation provided. Why it is safe to use 1-D results against 2-D ones?
Round 3
Reviewer 1 Report
After correcting a small typo in line 222, i.e. '...Trifunac (no Trifullac) and Brady [32]..', the paper can be moved to production. No further review is required.
